# Hydraulic Characteristics and Vortex Characteristics of the Flow around the Piped Vehicle with Different Diameter-to-Length Ratios

Lei Sun, Xihuan Sun * and Yongye Li

College of Water Resource Science and Engineering, Taiyuan University of Technology, Taiyuan 030024, China
* Correspondence: sunxihuan@tyut.edu.cn; Tel.: +86-135-1360-0012

**Abstract:** Hydraulic pipeline transportation of a piped vehicle is a new mode of transportation with energy-saving potential and environmental protection. In order to analyze the turbulent characteristics of the flow around the piped vehicle, a large eddy simulation (LES) method was adopted to simulate the hydraulic characteristics and vortex characteristics of the flow at a Reynolds number of $Re$ = 140,467 with diameter-to-length ratios of 0.4, 0.5, 0.6, and 0.7. The results showed that the main factor that affected the gap flow velocity, the backflow area length, and the turbulence intensity was the cylinder diameter in the diameter-to-length ratio of the piped vehicle. The backflow area lengths for piped vehicles with different diameter-to-length ratios were all less than 1 D, and the axial disturbance distances were about 7.5 D. In addition, a variety of vortex structures existed in the gap flow and the rear flow areas. At the beginning of vortex development, ring vortices were generated at the front and rear ends of the cylinder body. Subsequently, the front ring vortex fell off along the cylinder body and evolved into hairpin vortices. At the same time, a reflux vortex was formed after the rear ring vortex broke away from the cylinder body, and wake vortices were generated behind the rear supports. Finally, some worm vortices were dispersed from the wake vortices. These results can further improve the theoretical system for the hydraulic pipeline transportation of piped vehicles and can provide a theoretical basis for industrial application.

**Keywords:** piped vehicle; diameter-to-length ratio; hydraulic characteristics; vortex; large eddy simulation

## 1. Introduction

The development of hydraulic pipeline technology has ranged from the hydraulic slurry pipeline (which mixes solid particles with liquid) to the hydraulic coal log pipeline (which compacts solid particles together) and the hydraulic capsule pipeline (which stores raw materials in airtight containers). The range of transporting materials has gradually expanded, while the pre-processing procedures for materials have gradually reduced, and solid–liquid separation is now realized in the transportation process.

Critical velocity and drag loss characteristics are the main objects of study for hydraulic slurry pipeline: a too low transporting velocity leads to pipeline blockage, while a too high transporting velocity causes project investment to increase. Therefore, it is crucial to determine a reasonable velocity. Thus far, Durand, Wasp, Shook, Fei, and Zou have proposed representative critical velocity formulas, which are only applicable in their respective ranges due to the fact that these formulas have been obtained under certain experimental conditions with widely varying results. The key to the study of hydraulic slurry pipelines is drag loss calculation. There are many factors that affect the frictional resistance of slurry transportation, such as flow velocity, pipe wall material, the physical properties of material particles, the concentration of the slurry, the inner diameter of the pipe, etc. Durand's resistance loss calculation model based on gravity theory [1,2], Chen Wenguang's resistance loss calculation model [3,4], and the thick slurry transportation

resistance loss calculation model [5] are three representative slurry resistance calculation models at present.

In the 1990s, Liu [6] conducted hydraulic pipeline experiments on cylindrical coal logs, analyzing their transporting characteristics and economic feasibility at the University of Missouri, USA, and formally proposed the concept of a hydraulic coal log pipeline. A hydraulic coal log pipeline needs to compact bulk material into a cylindrical shape before transportation. Brett et al. [7] obtained the optimal moisture content for the rapid compaction of coal columns, which provided a theoretical reference for pre-processing. Zhang and Zhao et al. [8,9] performed an experimental investigation of the pressure distribution on a coal log's surface while in constant flow in a pipe. Lin [10] studied the starting speed of a coal log in a pipeline through theoretical derivation. Li [11] designed an experiment to study the moving speed of a coal log in a pipeline.

Compared with the hydraulic slurry pipeline and the hydraulic coal log pipeline, the most important feature of the hydraulic capsule pipeline is that the material to be transported is separated from the liquid by a carrier, thus reducing contamination of the liquid. Bageni et al. [12,13] analyzed the forces acting on a capsule with experimental studies and finite element analysis methods. Ulusarlan et al. [14] experimentally established an equation of friction coefficient for the two-phase flow of capsule water in different flow states and found that the capsule inevitably collided and rubbed against the pipe wall during transportation. Therefore, Sun [15] improved the structure of the capsule body so that the capsule, which was named the "piped vehicle", always maintained a coaxial line with the pipe during transportation, avoiding the large area collision with the pipe wall. Then, the starting conditions and the moving speed characteristics of different types of piped vehicles were discussed by Sun et al. [16,17] through model experiments. Li et al. [18] combined model experiments and theoretical analysis to study the energy consumption of the piped vehicle hydraulic transportation and deduced formulas for calculating the total energy consumption that were validated experimentally.

In the process of the hydraulic pipeline transportation of a piped vehicle, a fluid–structure coupling problem exists between the piped vehicle and the fluid around the piped vehicle. The load of water flow acting on the piped vehicle causes deformation of the piped vehicle, and the deformation of the piped vehicle in turn acts on the water flow. With the development of hydraulic systems toward high-pressure and high-power weight ratios, the fluid–structure coupling vibration of hydraulic pipeline systems and the failure problems caused by vibration are becoming increasingly serious and have become a bottleneck restricting the development of hydraulic pipeline systems. Therefore, scholars have conducted much research on the vibration characteristics of hydraulic pipeline systems. Cao [19] analyzed the vibration characteristics of a marine hydraulic pipeline system, and his results were of theoretical value and practical significance for the safe operation of ships. Quan et al. [20] studied the axial vibration response characteristics of fluid–structure interaction vibration for an aircraft hydraulic pipe considering friction coupling. Gong et al. [21] designed a new type of MR damping clamp structure to solve the problem of low-frequency vibration for a hydraulic pipeline, and its performance was simulated and tested. Khudayarov and Turaev [22] developed a computational algorithm for solving the vibration problems of composite pipelines conveying pulsating liquid and studied the stability and amplitude time characteristics of the vibration of composite pipelines with pulsating fluid flowing in them. Gao et al. [23] provided a detailed review of current vibration control technologies in hydraulic pipeline systems and gave some suggestions for the application of vibration control technologies in the engineering field. Zhang et al. [24,25] conducted numerical simulations of fluid–structure coupling between the structural response of a piped vehicle and the internal flow field to further analyze the hydraulic characteristics of the vibration motion of a piped vehicle in a flat, straight pipe. Yang [26] and Jia [27] carried out a study on the dynamic characteristics of a piped vehicle during vibration motion that laid a theoretical basis for the safe operation of a hydraulic pipeline transportation system for piped vehicle.

A vortex is a special form of fluid motion that is derived from the rotation of fluid elements. This organized structure ranges from spiral galaxies in the universe to coherent structures in turbulent currents and is widely present in nature and society [28]. Vortices are the "muscles" in viscous flows and have the function of transporting or pumping fluid mass and momentum. Due to the rapid development of experimental visualization methods and techniques, as well as modern computing hardware technology and CFD techniques, in recent years, both experimental [29–31] and numerical simulation researches [32–35] on vortices have entered a new stage.

The structure of the transport carrier (a piped vehicle) in a hydraulic piped vehicle pipeline system is a complex, intersecting cylindrical system, and the flow around the piped vehicle at high Reynolds numbers is a form of turbulent and complex multiscale vortex motions in space and time. These vortices with different scales and strengths are constantly moving, deforming, interacting, and finally, dispersing into small-scale coherent structures which then dissipate into heat, resulting in the energy consumption of the system and affecting the efficiency of transportation. There have been some important research results [36–38] on the vortex characteristics of flow around cylindrical objects. However, there has been little research on the vortex characteristics of complex cylindrical system, such as a piped vehicle. Therefore, this paper adopts the 3D large eddy simulation (LES) method to numerically analyze the hydraulic characteristics and vortex motions of flow around a piped vehicle at a high Reynolds number.

## 2. Numerical Simulation Method

### 2.1. Geometric Model Setup and Meshing

The geometric model of the piped vehicle was a cylinder body with six supports that were evenly distributed at angles of 120 degrees on both ends of the cylinder. In this paper, four models of piped vehicles were selected. The lengths of the cylinder bodies were 100 mm and 150 mm, and the diameters were 60 mm and 70 mm; the diameter-to-length ratios of the four models were 0.4, 0.5, 0.6, and 0.7. To reduce the numerical calculation cycle and save computational resources, the sizes of the supports for the different piped vehicles were consistent. The geometric model of the piped vehicle is shown in Figure 1. The pipeline model was a smooth, circular pipe with a diameter of D = 0.1 m, and D was used as the characteristic length for the studies in this paper. In order to eliminate the influences of the inlet and outlet boundaries on the numerical simulation results, it was preferred to choose a longer straight pipe model [39]. The length of the simulated pipe was 2 m. Along the flow direction, the left was the front end of the piped vehicle, while the right was the rear end, so the inlet was 0.6 m from the front end, and the outlet was 1.3 m or 1.25 m from the back end. The calculation domain is shown in Figure 2. A geometric 3D model was created using AutoCAD and then generated as a SAT file, which stored the ASCII text format.

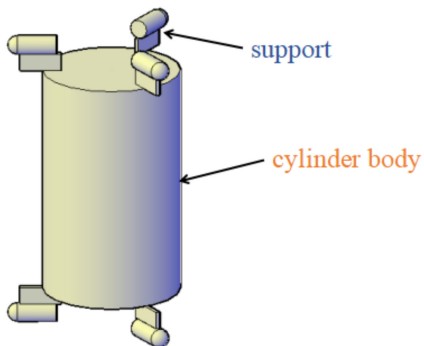

**Figure 1.** Geometric model of the piped vehicle.

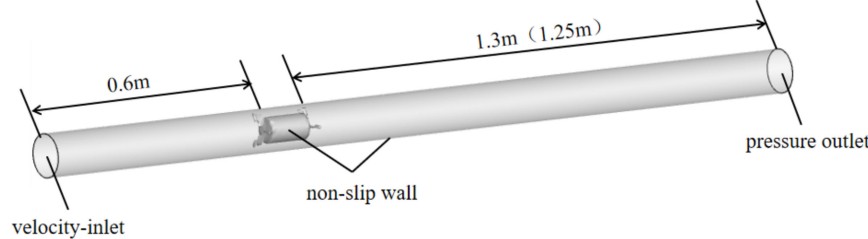

**Figure 2.** Computational domain and boundary conditions.

ICEM software was used to divide the calculation area into nonstructural tetrahedral meshes, and the local mesh was refined for the annular gap field and the field behind the piped vehicle, as shown in Figure 3. The mesh irrelevance was verified firstly to find a more suitable point between the calculation accuracy and the calculation overhead, and mesh refinement was adopted in a 2-fold encryption method. Three grid sizes of 1.5 mm, 2 mm, and 2.5 mm were chosen for comparison. The vorticity of the center point in the cross-section, a characteristic length from the end of the cylinder body at the moment $t = T$, was used as the verification parameter. The results are shown in Table 1. It can be seen that the calculation results of the 2.5 mm grid were different from those of 1.5 mm and 2 mm grids, and the vorticity values were less sensitive for the 1.5 mm and 2 mm grids. Therefore, under the premise of ensuring calculation accuracy and efficiency, the 2 mm grid was chosen in this paper.

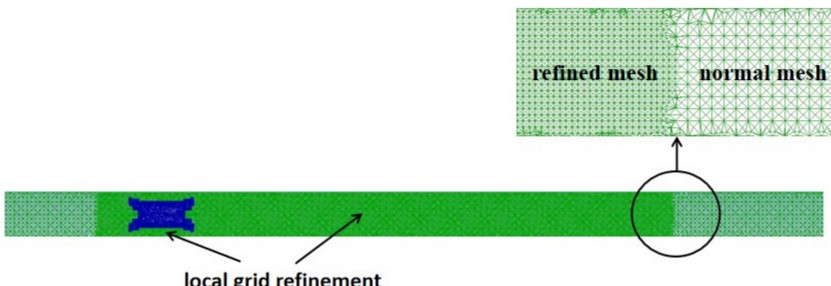

**Figure 3.** Mesh settings.

**Table 1.** Total number of grids and numerical calculation values for the three grid sizes.

| $K$ | Total Number of Grids | | | Vorticity (1/s) | | |
|---|---|---|---|---|---|---|
| | **2.5 mm** | **2 mm** | **1.5 mm** | **2.5 mm** | **2 mm** | **1.5 mm** |
| 0.4 | 15,083,139 | 30,102,897 | 58,471,317 | 421.895 | 455.026 | 453.876 |
| 0.5 | 14,952,093 | 29,831,097 | 58,194,097 | 394.284 | 438.463 | 436.998 |
| 0.6 | 15,200,257 | 30,340,955 | 58,949,370 | 297.387 | 331.485 | 334.275 |
| 0.7 | 15,133,129 | 30,158,341 | 58,606,148 | 374.473 | 398.799 | 390.648 |

*2.2. Boundary Conditions and Governing Equations*

The inlet boundary was set as the velocity inlet. Because the viscous sublayer and transition layer in the velocity distribution of the pipe flow were very small, the logarithmic profile obtained from Nikuradse's experiment was used to define the velocity inlet, which was realized by "User Defined" according to Equations (1)–(4).

$$\frac{u}{u_*} = 5.75\lg\frac{u_* y}{\nu} + 5.5 \tag{1}$$

$$u_* = \sqrt{\frac{\lambda}{8}} U_{in} \tag{2}$$

$$\lambda = 0.0032 + \frac{0.221}{Re^{0.237}} \ (Re < 10^6) \tag{3}$$

$$Re = \frac{U_{in}D}{\nu} \qquad (4)$$

where $u$ is the velocity at a point in the inlet, $y$ is the radial distance from the point to the pipe wall, $u_*$ is the friction velocity, $\lambda$ is the friction coefficient, $U_{in}$ is the average velocity, $\nu$ is the kinematic viscosity, and $D$ is the pipe diameter. The turbulent condition of the inlet was set to "no perturbation", i.e., the turbulent intensity was 0. The fluid was water with a density of 998.2 kg/m$^3$, and the kinetic viscosity was $1.005 \times 10^{-3}$ Pa·s when the temperature was 20 °C. The outlet boundary was set as the pressure outlet, and the pressure value was set to 7840 pa, which was measured by experiments. The walls of both the pipe and the piped vehicle were defined as fixed walls without slip (non-slip wall). The roughness height of the walls were set to 0.01 mm, which was consistent with the roughness of the plexiglass used in the experiments. The boundary conditions are shown in Figure 2.

The Reynolds number was 140,467, which meant that the flow around the piped vehicle was three-dimensional, incompressible, and unstable turbulence. For three-dimensional incompressible viscous fluid, the tensor form of the equation of motion can be expressed as follows.

$$\frac{\partial u_i}{\partial t} + u_j \frac{\partial u_i}{\partial x_j} = -\frac{1}{\rho}\frac{\partial p}{\partial x_i} + \nu \frac{\partial^2 u_i}{\partial x_j x_j} \qquad (5)$$

where $t$ is the time, $p$ is the static pressure, $\rho$ is the density, and $\nu$ is the kinematic viscosity.

Large eddy simulation is a method for the direct calculation of large-scale eddies, which play an important role in flow, and the Reynolds-averaged calculation of small-scale eddies, which play smaller roles in flow. The process includes filtering small-scale pulsations using filter functions to obtain the control equations for large-scale motion, introducing subgrid-scale additional stress terms into the equation, and solving the N–S equation. Thus, the filtered N–S equation and the continuity equation are the control equations for large eddy simulation.

$$\frac{\partial u_i}{\partial t} + \frac{\partial}{\partial x_j}\left(\overline{u_i}\,\overline{u_j}\right) = -\frac{1}{\rho}\frac{\partial \overline{p}}{\partial x_i} - \nu \frac{\partial^2 \overline{u_i}}{\partial x_j x_j} - \frac{\partial \tau_{ij}}{\partial x_j} \qquad (6)$$

$$\frac{\partial \overline{u_i}}{\partial x_i} = 0 \qquad (7)$$

where $u_i$ and $u_j$ are the filtering velocities in the x direction and y direction, respectively. Equation (6) was obtained by filtering the N–S equation (Equation (5)) with a filter function, and a new, unknown quantity, the subgrid stress $\tau_{ij}$, appeared in Equation (6). In order to make the control equations closed, further construction of a subgrid-scale model was required. The subgrid stress is defined as follows.

$$\tau_{ij} = \overline{u_i u_j} - \overline{u_i}\,\overline{u_j} \qquad (8)$$

Smagorinsky [40] proposed using the Boussinesq hypothesis to calculate the subgrid stress, i.e.,

$$\tau_{ij} - \frac{1}{3}\tau_{kk}\,\delta_{ij} = -2\mu_t \overline{S_{ij}} \qquad (9)$$

where $\mu_t$ is the subgrid turbulent viscosity, and $\overline{S_{ij}}$ is the strain rate tensor defined as follows.

$$\overline{S_{ij}} = \frac{1}{2}\left(\frac{\partial \overline{u_i}}{\partial x_j} + \frac{\partial \overline{u_j}}{\partial x_j}\right) \qquad (10)$$

In this paper, we adopted an LES method based on a WALE (wall-adapting local eddy viscosity) subgrid-scale model [41] in which the eddy viscosity coefficient near the wall was proportional to the third power of the vertical distance from the wall, which could

accurately reflect the characteristics of turbulent motion in the near-wall region. Turbulent viscosity $\mu_t$ can be expressed as follows.

$$\mu_t = \rho L_s^2 \frac{\left(S_{ij}^d S_{ij}^d\right)^{3/2}}{\left(\overline{S}_{ij}\overline{S}_{ij}\right)^{5/2} + \left(S_{ij}^d S_{ij}^d\right)^{5/4}} \tag{11}$$

$$L_s = min\left(kd, C_w V^{1/3}\right) \tag{12}$$

$$S_{ij}^d = \frac{1}{2}\left(\overline{g}_{ij}^2 + \overline{g}_{ji}^2\right) - \frac{1}{3}\delta_{ij}\overline{g}_{kk}^2 \tag{13}$$

$$\overline{g}_{ij} = \frac{\partial \overline{u}_i}{\partial x_j} \tag{14}$$

where $L_s$ is the mixing length at the subgrid-scale and $k$ is the von Kármán constant. Intensive validation has shown consistently superior results with $C_w = 0.325$, so the coefficient $C_w$ was set to 0.325.

*2.3. Solving Method and Discrete Method*

The finite volume method was used to solve the control equations, and its solution strategy was to calculate the variables at control points using the fluxes on boundary surfaces or lines. The SIMPLEC [42,43] algorithm of the separated solver was chosen to solve the pressure–velocity coupled equations. SIMPLIC is an improvement on SIMPLE. The pressure correction under-relaxation factor was set to 1.0 in SIMPLIC, which aided in convergence speed-up. The spatial discretization was in a bounded central differencing formulation, and the time was discretized in a bounded second-order implicit transient formulation. The dimensionless time step is defined as follows.

$$T = \frac{U_{in}\Delta t}{D}, \tag{15}$$

In order to ensure that the time step was set reasonably, it was necessary to ensure that the Courant number was less than 1, i.e.,

$$\frac{\Delta t}{\Delta x / u} < 1 \tag{16}$$

where $\Delta t$ is the time step, $u$ is the velocity (here, $u$ was the average velocity of the pipe inlet $U_{in}$), and $\Delta x$ is the minimum grid size. Thus, the dimensionless time step was 0.0014. The number of iterations per time step was set to 20, and the residual was set to $10^{-6}$.

## 3. Experimental Procedure

### 3.1. Experimental System

In order to verify the rationality and applicability of the model and algorithm, a series of physical experiments were conducted in this paper. As shown in Figure 4, the experimental platform mainly consisted of a round, plexiglass pipe with an inner diameter of 100 mm and a thickness of 5 mm, a water tank, a centrifugal pump, a regulator valve, and an electromagnetic flow meter. The pipe was made up of two flat, straight sections and one flat, curved section; the total length was 26.8 m. The water in the pipe was powered by a centrifugal pump with a rated power of 11 KW and a speed of 2900 r·min$^{-1}$, which could provide a maximum flow rate of 110 m$^3$/h. The flow rate in this experiment was 40 m$^3$/h. The flow rate was controlled by the regulator valve and the electromagnetic flow meter. The electromagnetic flow meter was installed at a distance of 1.3 m from the inlet elbow. The test section was located downstream from the flat, curved section at 4.8 m from the pipe outlet. Particle image velocimetry (PIV) was used to measure the three-dimensional velocities of the flow around the piped vehicle. In order to reduce the light refraction of

the pipe wall, a rectangular water jacket was set outside the pipe over the test section, as shown in Figure 5.

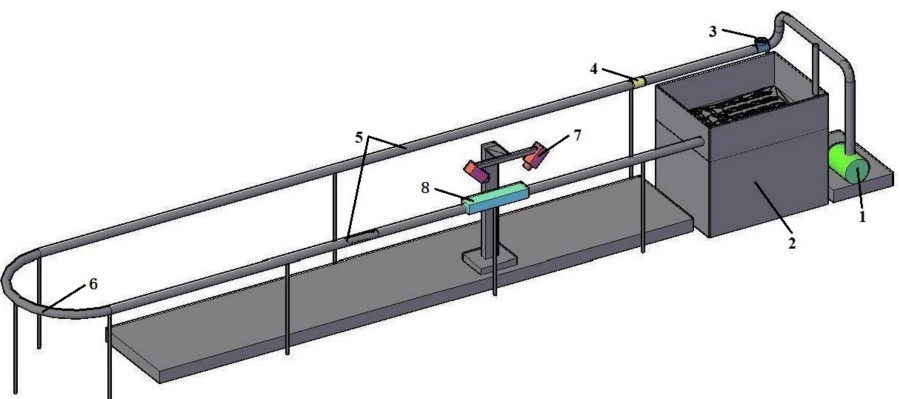

**Figure 4.** Diagram of experimental platform layout: 1—centrifugal pump; 2—tank; 3—regulator valve; 4—electromagnetic flow meter; 5—flat, straight pipe; 6—flat, curved pipe; 7—particle image velocimeter; 8—test section.

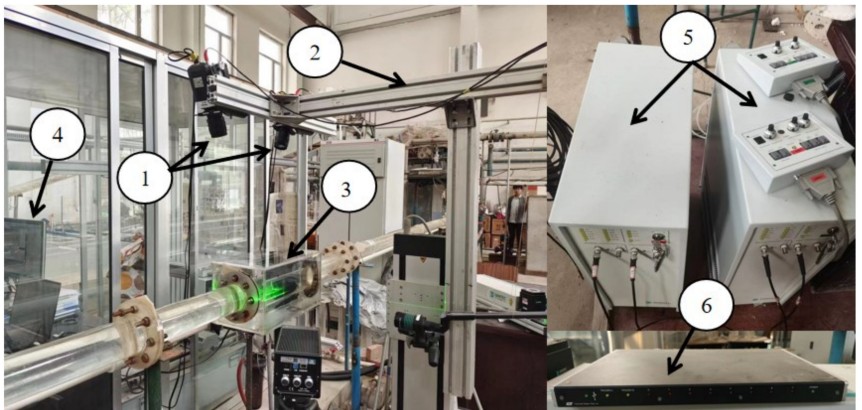

**Figure 5.** Particle image velocimetry system: 1—CCD camera; 2—coordinate frame; 3—square water jacket; 4—DynamicStudio software; 5—laser; 6—synchronizer.

### 3.2. Selection of Cross-Section and Layout of Measurement Points

The gap flow and the rear flow of two models with diameter-to-length ratios of 0.5 and 0.6 were verified with experiments. For gap flow, the central cross-section perpendicular to the z-axis was taken as the test section, and the radial line from the cylinder body wall to the pipe wall along the negative direction of the y-axis (one support was arranged along the positive direction of the y-axis) was taken as the test line. The measurement points were arranged at 2 mm intervals along the test line. For rear flow, the center axis of the pipe along the z-axis was taken as the test line, and the measurement points were set at 20 mm intervals along the test line starting from the back end of the cylinder body. The test section, the test lines, and the measurement points were arranged as shown in Figure 6.

### 3.3. Validation of Simulated Results

The experimental and simulated results of the three-dimensional velocities of the gap flow and rear flow for piped vehicles of $Re$ = 140,467 with diameter-to-length ratios of 0.5 and 0.6 are given in Figures 7 and 8 in dimensionless coordinates. The horizontal coordinate in Figure 6 is the ratio of the distance ($s$) from the measurement point to the outer wall of the cylinder body to the gap width ($B$). The horizontal coordinate in Figure 7 is the ratio of the distance ($S$) from the measurement point to the rear of the cylinder body to the pipe diameter ($D$). The vertical coordinates of Figures 6 and 7 are all the ratios of the

axial ($u_z$), radial ($u_y$), and circumferential ($u_x$) time-averaged velocities of the measurement point to the average velocity of the pipe inlet ($U_{in}$).

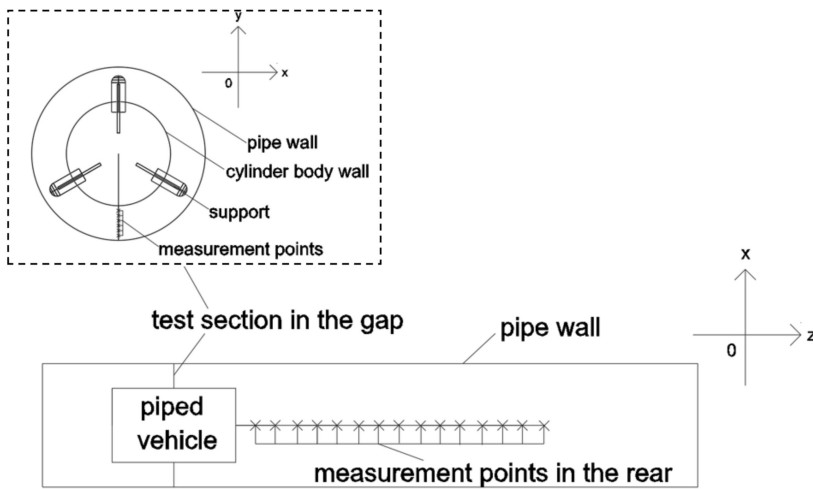

**Figure 6.** Layout of the test section, test lines, and measurement points for gap flow and rear flow.

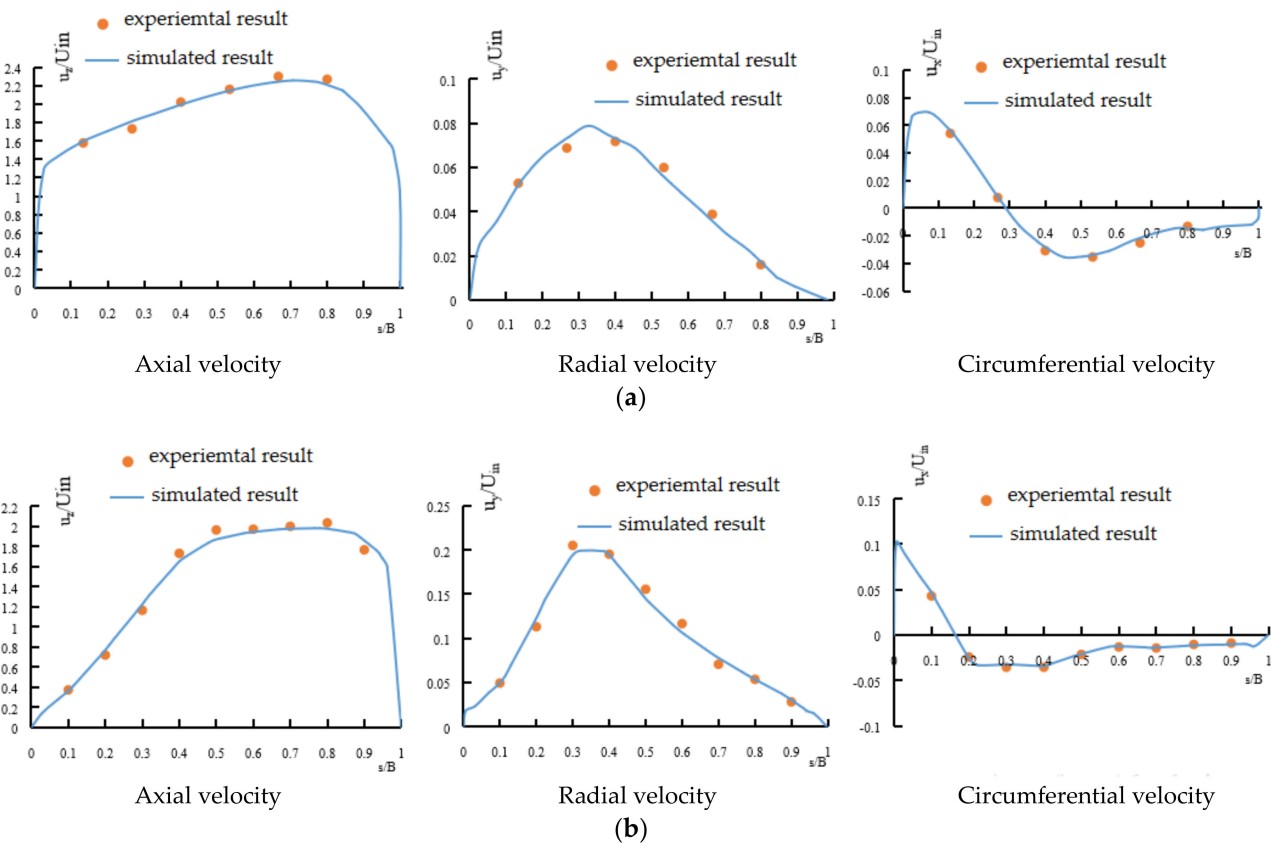

**Figure 7.** The experimental and simulated results of the three-dimensional velocities for gap flow: (**a**) $K = 0.5$; (**b**) $K = 0.6$.

It can be seen from the above figures that the numerical simulations were consistent with the experimental results. The maximum relative errors of the gap flow velocity and the rear flow velocity were 4.32% and 3.85%, respectively, both of which did not exceed 5%, so the choice of the numerical simulation scheme used in this paper was considered reasonable. The errors could be due to the difference in the time step used to obtain instantaneous

velocity between the numerical simulations and the physical experiments. The time step was 0.0001 s in the numerical simulations while it was 0.025 s in the experiments.

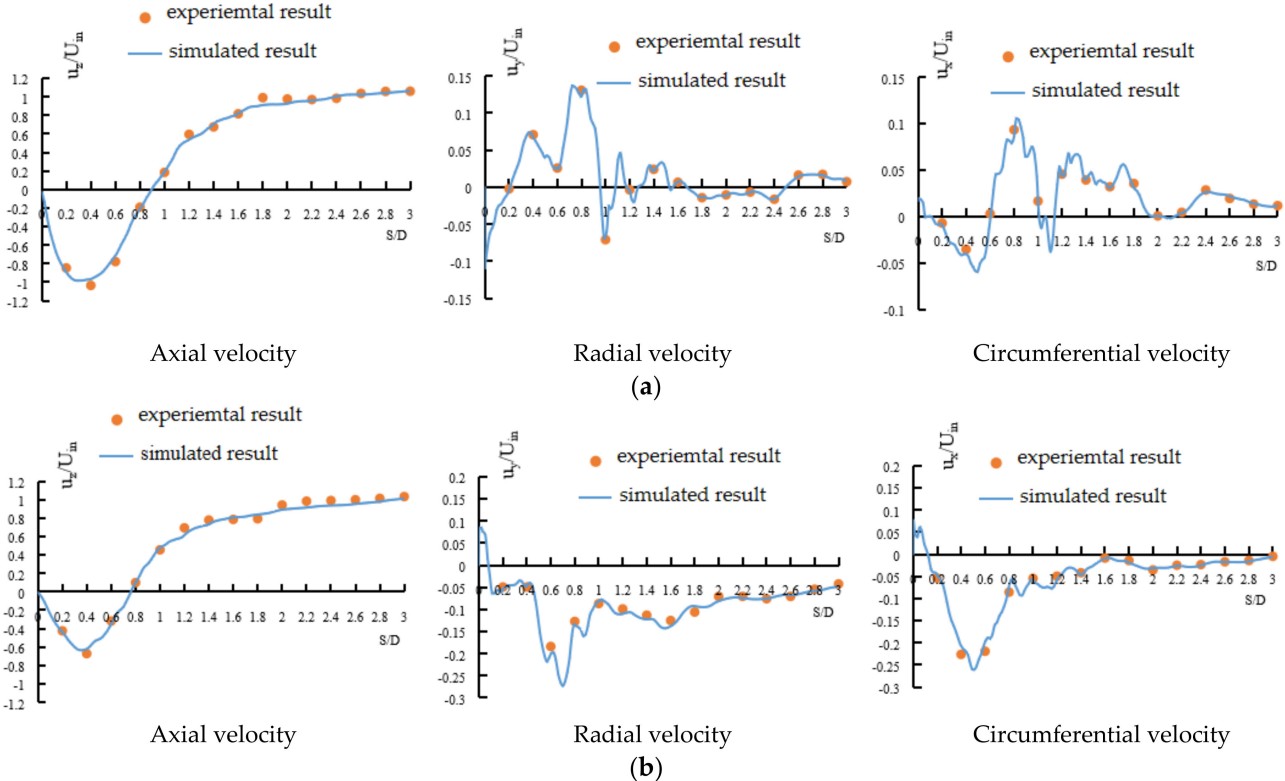

**Figure 8.** The experimental and simulated results of the three-dimensional velocities for rear flow: (**a**) $K = 0.5$; (**b**) $K = 0.6$.

The magnitude of the axial velocity for gap flow was 1–2 orders of magnitude larger than those in both the radial and circumferential directions, indicating that axial velocity dominated in the gap flow. The velocities in the radial and circumferential directions could reach the same order of magnitude as the axial velocity due to the high mixing of the turbulent flow in the rear flow. The axial velocity in the rear flow first decreased and then increased along the flow direction.

## 4. Results and Discussion

### 4.1. Time-Averaged Velocity Distribution

The dimensionless, time-averaged velocity distribution clouds in the XZ plane for piped vehicles with different diameter-to-length ratios were obtained by the statistical function of the large eddy simulation calculation. As shown in Figure 9, the time-averaged velocity in the inlet of the gap flow increased abruptly with a large velocity gradient, while a large low-velocity area appeared in the rear flow where the velocity gradient was also large. Therefore, it was concluded that there must be vortices of greater strength in these two places. By comparing time-averaged velocity distribution clouds for piped vehicles with different diameter-to-length ratios in the XZ plane, it was found that if the cylinder length was constant, when the water met the front end of the piped vehicle, the obstructed area increased with the increase in diameter-to-length ratio. Thus, the area allowed for passing became smaller, so the time-averaged velocity of the gap flow must increase under the condition of a constant inlet flow rate. If the cylinder diameter was constant, the time-averaged velocity did not change much with an increase in the diameter-to-length ratio because the maximum value of the time-averaged velocity appeared at the inlet of the gap flow where the streamline changed abruptly, and a change in the cylinder length did not affect the shape and size of the inlet of the gap flow.

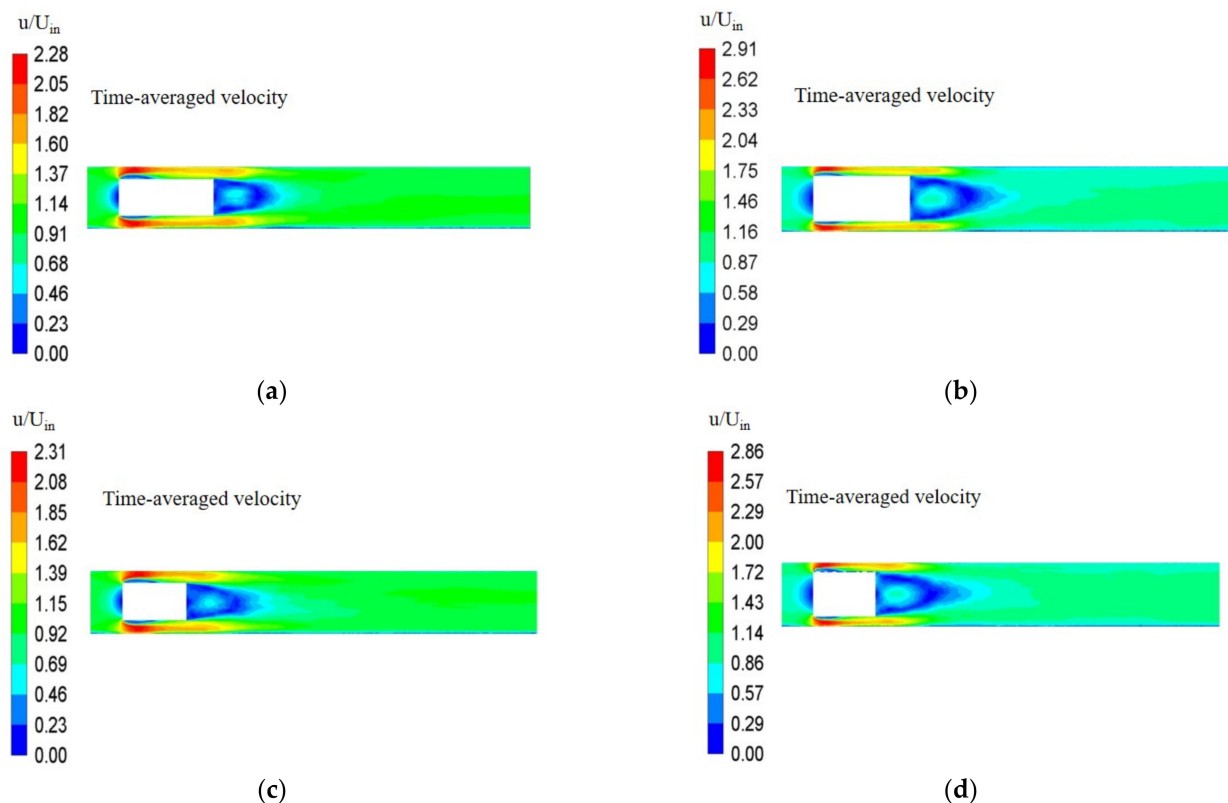

**Figure 9.** Dimensionless, time-averaged velocity distribution cloud in XZ plane: (**a**) $K = 0.4$; (**b**) $K = 0.5$; (**c**) $K = 0.6$; (**d**) $K = 0.7$.

*4.2. Backflow Area Length*

The backflow area behind the piped vehicle was defined as the area in which the direction of the time-averaged downstream velocity was opposite to the average velocity of the pipe inlet, i.e., $\frac{u_z}{U_{in}} < 0$. Its length was recorded as $S_r$. The dimensionless, time-averaged downstream velocity distribution cloud in the XZ plane and the dimensionless, time-averaged downstream velocity variation curve along the center line of the pipe are plotted in Figure 10. From the figure, it can be seen that when the water flowed past the rear end of a piped vehicle, a low-velocity area with annular distribution was formed behind the piped vehicle in which the downstream velocity grew smaller from the outer ring to the inner ring. The fluid formed a closed loop, creating a "dead zone" in which the downstream velocity of the center was the smallest. Thus, the distance from the center of the rear end of the cylinder body to the tail end of the "dead zone" was defined as the length of the backflow area. The lengths of backflow areas for pipe vehicles with different diameter-to-length ratios are listed in Table 2. It can be seen that the lengths of the four diameter-to-length ratios were all less than 1 D. When the cylinder length was constant and the diameter-to-length ratio became larger, the backflow area became larger; when the cylinder diameter was constant and the diameter-to-length ratio increased, the backflow area was almost constant. This indicates that the main factor affecting the size of the backflow area was the value of the diameter in the diameter-to-length ratio.

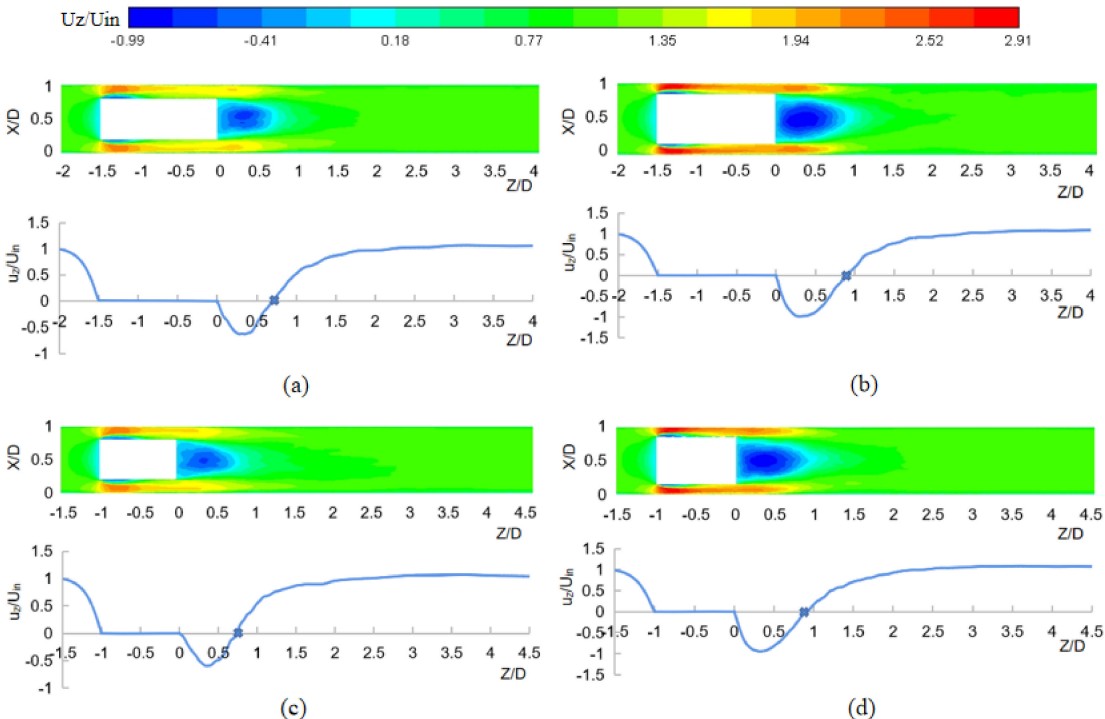

**Figure 10.** Dimensionless, time-averaged downstream velocity distribution cloud in XZ plane and dimensionless, time-averaged downstream velocity variation curve along center line of pipe: (**a**) $K = 0.4$; (**b**) $K = 0.5$; (**c**) $K = 0.6$; (**d**) $K = 0.7$.

**Table 2.** Backflow area lengths for piped vehicles with different diameter-to-length ratios.

| $K$ | $S_r$ |
|---|---|
| 0.4 | 0.705 D |
| 0.5 | 0.895 D |
| 0.6 | 0.757 D |
| 0.7 | 0.901 D |

### 4.3. Vorticity Variation Characteristics

In 1858, Helmholtz [44] introduced the concepts of vortex line, vortex tube, and vortex filament, as well as Helmholtz's three laws, defining vorticity as the curl of the velocity vector with the following mathematical expression.

$$\Omega = \nabla \times u \tag{17}$$

where $\nabla = i\frac{\partial}{\partial x} + j\frac{\partial}{\partial y} + k\frac{\partial}{\partial z}$ is the Hamiltonian operator, expanded by the law of the product of two vector forks. Then, the vorticity can be calculated as follows.

$$\Omega = \nabla \times u = \left(\frac{\partial u_z}{\partial y} - \frac{\partial u_y}{\partial z}\right)i + \left(\frac{\partial u_x}{\partial z} - \frac{\partial u_z}{\partial x}\right)j + \left(\frac{\partial u_y}{\partial x} - \frac{\partial u_x}{\partial y}\right)k \tag{18}$$

Figure 11 shows the results of the vorticity calculations in the XZ cross-section at different moments for piped vehicles with diameter-to-length ratios of 0.5 and 0.6. The vorticity distribution confirmed that the vortices in the flow around a piped vehicle were mainly generated in the gap flow area and the rear flow area behind the piped vehicle.

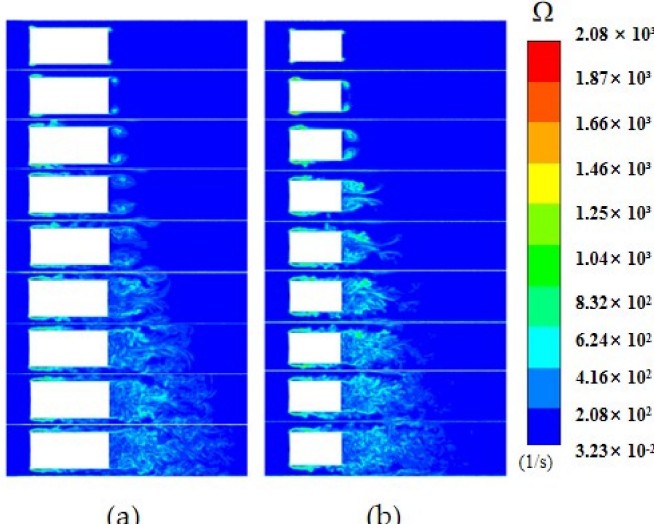

**Figure 11.** Vorticity distribution at different moments: (**a**) $K = 0.5$; (**b**) $K = 0.6$.

As can be seen from the development of the vorticity distribution, the locations where the vorticity first appeared were at the front and rear ends of the cylinder body. Afterward, a region with large vorticity appeared near the cylinder body wall at the front of the annular gap, and it can be seen from Figure 8 that the velocity gradient in this region was large. This was due to the fact that after the water flowed past the front end of the cylinder body, a backflow was generated due to the obstruction effect of the piped vehicle, which resulted in the separation of the boundary layer on the wall of the cylinder body. Therefore, the velocity gradient was larger in this region.

At this time, a pair of vortices that rotated in opposite directions and had symmetric structures appeared behind the piped vehicle. This was because when the water flowed through the gap and past the rear end, a part of the water retained axial momentum to continue to flow backwards, while the other part of the water flowed around the rear end of the cylinder body along all the radial directions. The water on the top and bottom sides had a radial velocity in the same direction and of the same order of magnitude as the axial velocity. It also had a backward axial velocity, and thus, this pair of vortices continuously developed and expanded with time both backward and toward the center of the pipe. It can also be seen from Figure 10 that there was a place located near the shear layer distributed along the cylinder edge behind the piped vehicle where the vorticity value was large. This was because the velocity near the shear layer had a large gradient in the radial direction, so the transient vorticity was more obvious. This was the vortex generated by the Kelvin–Helmholtz instability [45].

*4.4. Vortex Structure Characteristics*

Hunt [46] believed that vortex motion existed when the rotational part of a velocity gradient tensor was larger than the deformation part and that vortex structure was identified by the isosurface, where the value of $Q$ was positive. $Q$ can be calculated as follows.

$$Q = \frac{1}{2}\left(\|\omega\|^2 - \|S\|^2\right) \tag{19}$$

where $\omega$ denotes the rotational tensor, and $S$ denotes the shear deformation tensor.

Figure 12 shows the three-dimensional instantaneous vortex structure contours around a piped vehicle with a diameter-to-length ratio of 0.6. The contours were colored by the vorticity and clearly showed the development characteristics of vortices around the piped vehicle.

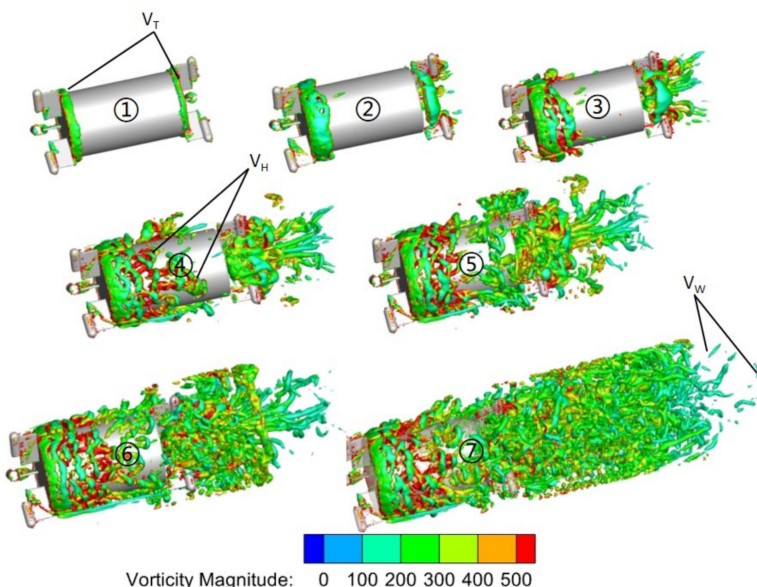

**Figure 12.** Three-dimensional instantaneous vortex structure isosurface around piped vehicle.

In the gap flow, at the beginning of vortex development, a ring vortex ($V_T$) was generated at the front end of the cylinder body. Then, the ring vortex gradually thickened and fell off along the cylinder body and evolved into hairpin vortices ($V_H$) that were tilted outward. This was because when the flow encountered the piped vehicle, the velocity of the flow decreased and the pressure increased. An adverse pressure gradient was generated at the front of the gap, and the adverse pressure gradient was large enough to form a backflow. The interaction between the backflow and the incoming flow and the interaction between the backflow and the wall of the cylinder body caused the formation of the hairpin vortices. The hairpin vortices gradually increased. In this process, tubular vortices that were generated by the three front supports gradually extended along the flow direction to the whole gap.

The vortices behind the piped vehicle could be divided into two parts: the reflux vortex formed by the backflow and the wake vortices generated by the rear supports. The reflux vortex evolved from the ring vortex which was generated by the rear end. After breaking from the cylinder body, the ring vortex was squeezed by the flow from the gap, which converged toward the center of the pipe, and developed inward along the radial direction. Then, a backflow was generated and a reflux vortex was formed. Meanwhile, the vortices generated by the rear supports were carried by the main stream, resulting in axial and radial motions. Therefore, three groups of wake vortices that developed outward obliquely were found behind the three rear supports. The wake vortices crossed and twisted. Subsequently, the vortices behind the piped vehicle were scattered as a number of disordered worm vortices ($V_W$) due to mutual entanglement, induction, and deformation.

*4.5. Vortex Formation Length*

The turbulence pulsations in the pipe increased due to the presence of the piped vehicle. The location where a turbulence pulsation peak occurred is the location of a fully formed vortex [47], and the location of a maximum value of the dimensionless pulsation amplitude is usually defined as the length of the vortex formation. Therefore, the vortex formation length is the length of the location of the maximum turbulence intensity.

$$I_{umax} = \frac{\sigma_{max}}{U} \tag{20}$$

where $\sigma_{max}$ is the maximum value of the root mean square value of fluctuating velocity, and $U$ is the time-averaged characteristic velocity. In this paper, $U$ was taken as the average velocity of the section far from the piped vehicle. Therefore, the vortex formation length

in the gap flow was the axis distance from the gap inlet to the location of the maximum turbulence intensity of the gap flow, and the vortex formation length in the rear flow was the axis distance from the rear end of the cylinder body to the location of the maximum turbulence intensity of the rear flow.

Figure 13 shows the variation trend of the cross-sectional average turbulence intensity along the flow direction. It can be seen that the maximum cross-sectional turbulence intensities of the gap flow for piped vehicles with diameter-to-length ratios of 0.4, 0.5, 0.6, and 0.7 were 15.42%, 18.49%, 14.39%, and 18.01%, respectively, and the maximum cross-sectional turbulence intensities of the rear flow were 23.29%, 42.11%, 21.65%, and 41.71%, respectively. We drew the conclusion that when the cylinder length was constant, the turbulence intensities of both the gap flow and the rear flow were proportional to the diameter-to-length ratio. However, when the cylinder diameter was constant, the turbulence intensities of both the gap flow and the rear flow changed little with change in the diameter-to-length ratio.

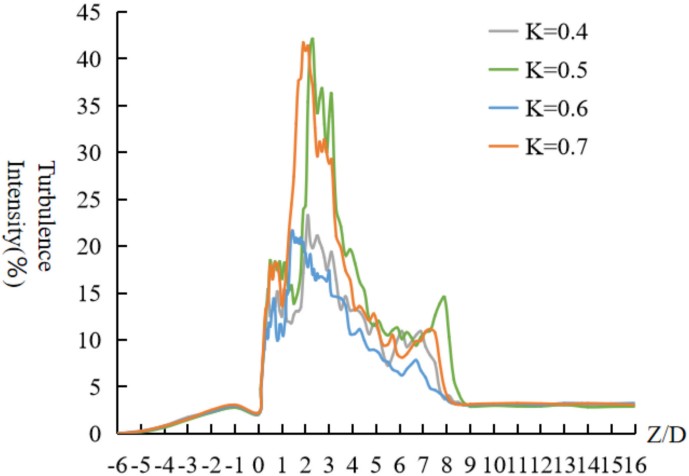

**Figure 13.** Variation trend of cross-sectional average turbulence intensity along the flow direction.

The vortex formation lengths for pipe vehicles with different diameter-to-length ratios are listed in Table 3. In comparison, it was found that if the diameter-to-length ratio changed, the vortex formation lengths of the gap flow changed slightly, while that of the rear flow changed obviously. The range of turbulence intensity fluctuation was the area where the flow was disturbed by the piped vehicle. It can be seen from Figure 13 that the gap flow was disturbed throughout. The disturbance range behind the piped vehicle extended to about a 7.5 D distance from the rear end, after which the water flow returned to a relatively steady turbulence state.

**Table 3.** Maximum cross-sectional average turbulence intensity and vortex formation lengths with different diameter-to-length ratios.

| K | 0.4 | | 0.5 | | 0.6 | | 0.7 | |
|---|---|---|---|---|---|---|---|---|
| | $I_{umax}$ (%) | L | $I_{umax}$ (%) | L | $I_{umax}$ (%) | L | $I_{umax}$ (%) | L |
| Gap flow | 15.42 | 0.4 D | 18.49 | 0.5 D | 14.39 | 0.6 D | 18.01 | 0.5 D |
| Rear flow | 23.29 | 0.6 D | 42.11 | 0.8 D | 21.65 | 0.4 D | 41.71 | 0.9 D |

## 5. Conclusions

We concluded the following:

(1)   The influences of the cylinder diameter on the gap flow velocity, the backflow area length, and the turbulence intensity were greater than those of the cylinder length.

When the cylinder length was constant, the gap flow velocity, the backflow area length, and the turbulence intensity were proportional to the diameter-to-length ratio.

(2) The effect of the diameter-to-length ratio on vortex formation length in the rear flow was greater than that in the gap flow. The presence of a piped vehicle made the original pipe turbulence disturbed, and the disturbance range extended to about a 7.5 D distance from the rear end.

(3) A variety of vortex structures existed in the gap flow and rear flow areas. At the beginning of vortex development, ring vortices were generated at the front and rear ends of the cylinder body, so the vorticity appeared at these two locations first. Immediately afterward, the front ring vortex thickened and fell off along the cylinder body, and evolved into hairpin vortices with larger values of vorticity attached near the cylinder body. At the same time, a reflux vortex was formed after the rear ring vortex broke away from the cylinder body, and three groups of cross-twisted wake vortices were generated behind the three rear supports. Finally, some worm vortices were dispersed from the wake vortices.

The hydraulic characteristics and vortex characteristics of the flow around a piped vehicle are studied in this paper, and the characteristics of the flow around two or more piped vehicles will be discussed in future works. These studies can provide a theoretical basis for the practical application and industrialization of the hydraulic pipeline transportation of piped vehicles.

**Author Contributions:** Formal analysis, L.S.; investigation, L.S. and Y.L.; data curation, L.S. and Y.L.; writing—original draft preparation, L.S.; writing—review and editing, X.S. and Y.L.; funding acquisition, X.S. and Y.L. All authors have read and agreed to the published version of the manuscript.

**Funding:** The research was funded by the National Natural Science Foundation of China (51179116, 51109155, 50579044) and the Natural Science Foundation of Shanxi Province (2015011067, 201701D221137).

**Institutional Review Board Statement:** Not applicable.

**Informed Consent Statement:** Not applicable.

**Data Availability Statement:** The data used to support the findings of this study are available from the corresponding authors upon request.

**Acknowledgments:** This research was supported by the Collaborative Innovation Center of New Technology of Water-Saving and Secure and Efficient Operation of Long-Distance Water Transfer Project at the Taiyuan University of Technology.

**Conflicts of Interest:** The authors declare no conflict of interest.

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
