# Peer review of "Hydraulic Characteristics and Vortex Characteristics of the Flow around the Piped Vehicle with Different Diameter-to-Length Ratios"

_water, doi:10.3390/w15010126_

Round 1

Reviewer 1 Report

This paper investigated the turbulent characteristics of the flow around the pipeline, it is meaningful and inspiring to the field of fluid dynamics of pipeline. The formulation, problem justification, simulations and visualization are clear and well prepared. However a few improvements are needed. The following revisions are recommended:

1. A concise and factual abstract is required. The abstract should state briefly the purpose of the research, the principal results and major conclusions.

2. The authors are suggested to add some latest references to give a better perspective of recent vibration characteristics of hydraulic pipeline system.

3. The boundary condition is not clear, please give the boundary condition of pipe for simulation.

4. In section 3, it is suggested to add the description of test bench. Such as pump speed, fluid pressure, sensor position, etc.

5. It is suggested to explain the error between the results of the simulation solution and experiment results in Figures 7-8..

6. The conclusion is too long to be shortened to specific points. Moreover, try to suggest some possible extensions to the work. This is very important and needs to be included in conclusions as it sets the correct premise for future directions.

7. There are some grammar and spelling mistakes inside the manuscripts. The authors are suggested to make a careful check again.

Reviewer 2 Report

The manuscript entitled „Hydraulic Characteristics and Vortex Characteristics of the Flow around the Piped Vehicle with Different Diameter-length Ratios“ utilized a three-dimensional Large Eddy Simulation method to achieve the numerical analysis of the hydraulic characteristics and vortex characteristics of the flow around the piped vehicle with different diameter-length ratios. The paper has provided a detailed description of the presented study and experiments. Results were also thoroughly analyzed. 

Below are some comments to improve the quality of this manuscript:

1.     The authors have stated that the Reynolds number Re=140467 was used in the presented study. Why do the authors choose this specific number? Please provide some illustrations.

2.     In the Section 2.1 (Figure 1), the authors have stated that, in the geometrical model, “the outlet was 1.3m or 1.25m from the back end”. So which distance did the author exactly use in the simulation? If this distance does not play an important role in this work, the authors should also explain it to prevent confusions.

3.     The quality (resolution) of Figure 3 should be improved. In the current form, the readers can hardly distinguish between the refined mesh and the normal mesh.

4.     In the Section 2.3, the authors have mentioned that the SIMPLEC algorithm was chosen to solve the equations. Please provide more details (reference or illustration) about this algorithm since only an abbreviation does not contain sufficient information.

5.     Many other studies have also used different CFD softwares (e.g. Flow-3D, ANSYS Fluent) to perform numerical vortex analysis. Following are some recommended references to be included as related work in the Introduction Section:

Sarkardeh, H., Reza Zarrati, A., Jabbari, E., & Marosi, M. (2014). Numerical simulation and analysis of flow in a reservoir in the presence of vortex. Engineering Applications of Computational Fluid Mechanics, 8(4), 598-608.

Huang, J., Sun, Y., Wang, T., Lueth, T. C., Liang, J., & Yang, X. (2020). Fluid-structure interaction hydrodynamics analysis on a deformed bionic flipper with non-uniformly distributed stiffness. IEEE Robotics and Automation Letters, 5(3), 4657-4662.

Round 2

Reviewer 1 Report

This paper has been revised according to comments. This paper can be accepted.

Reviewer 2 Report

The authors have revised the manuscript according to the reviewer's comments. Therefore, the reviewer agrees to publish the manuscript in the journal.